# Janus Kinase Inhibitors Improve Disease Activity and Patient-Reported Outcomes in Rheumatoid Arthritis: A Systematic Review and Meta-Analysis of 24,135 Patients

**DOI:** 10.3390/ijms23031246

**Published:** 2022-01-23

**Authors:** Lilla Tóth, Márk F. Juhász, László Szabó, Alan Abada, Fruzsina Kiss, Péter Hegyi, Nelli Farkas, György Nagy, Zsuzsanna Helyes

**Affiliations:** 1Department of Pharmacology and Pharmacotherapy, Medical School, University of Pécs, H-7624 Pécs, Hungary; drtothlilla@gmail.com (L.T.); k.fruzsina17@gmail.com (F.K.); zsuzsanna.helyes@aok.pte.hu (Z.H.); 2Institute for Translational Medicine, Szentágothai Research Centre, Medical School, University of Pécs, H-7624 Pécs, Hungary; flixjuhsz@gmail.com (M.F.J.); szabolaszlo1015@gmail.com (L.S.); alan.abada@gmail.com (A.A.); hegyi2009@gmail.com (P.H.); farkas.nelli@gmail.com (N.F.); 3Centre for Translational Medicine, Semmelweis University, H-1085 Budapest, Hungary; 4Heim Pál National Pediatric Institute, H-1089 Budapest, Hungary; 5Centre for Translational Medicine, Department of Medicine, University of Szeged, H-6725 Szeged, Hungary; 6Department of Anaesthesiology and Intensive Therapy, Medical School, University of Pécs, H-7624 Pécs, Hungary; 7Somogy County Kaposi Mór Teaching Hospital, H-7400 Kaposvár, Hungary; 8Division of Pancreatic Diseases, Heart and Vascular Center, Semmelweis University, H-1122 Budapest, Hungary; 9Institute of Bioanalysis, Medical School, University of Pécs, H-7624 Pécs, Hungary; 10Department of Rheumatology and Clinical Immunology, Department of Internal Medicine and Oncology, Semmelweis University, H-1027 Budapest, Hungary; 11Department of Genetics, Cell and Immunobiology, Semmelweis University, H-1089 Budapest, Hungary; 12Heart and Vascular Centre, Semmelweis University, H-1122 Budapest, Hungary; 13János Szentágothai Research Centre and Centre for Neuroscience, University of Pécs, H-7624 Pécs, Hungary

**Keywords:** rheumatoid arthritis, JAK inhibitors, analgesic effect, PROs, meta-analysis

## Abstract

Pain, fatigue, and physical activity are major determinants of life quality in rheumatoid arthritis (RA). Janus kinase (JAK) inhibitors have emerged as effective medications in RA and have been reported to exert direct analgesic effect in addition to reducing joint inflammation. This analysis aims to give an extensive summary of JAK inhibitors especially focusing on pain and patient reported outcomes (PRO). MEDLINE, CENTRAL, Embase, Scopus, and Web of Science databases were searched on the 26 October 2020, and 50 randomized controlled trials including 24,135 adult patients with active RA met the inclusion criteria. JAK inhibitors yielded significantly better results in all 36 outcomes compared to placebo. JAK monotherapy proved to be more effective than methotrexate in 9 out of 11 efficacy outcomes. In comparison to biological disease-modifying antirheumatic drugs, JAK inhibitors show statistical superiority in 13 of the 19 efficacy outcomes. Analgesic effect determined using the visual analogue scale and American College of Rheumatology (ACR) 20/50/70 response rates was significantly greater in the JAK group in all comparisons, and no significant difference regarding safety could be explored. This meta-analysis gives a comprehensive overview of JAK inhibitors and provides evidence for their superiority in improving PROs and disease activity indices in RA.

## 1. Introduction

Rheumatoid arthritis (RA) is a systemic autoimmune disease characterized by erosive inflammation of the joints, as well as systemic manifestations leading to pain and reduced quality of life [1]. The management of RA has substantially improved in the last few decades, particularly with the appearance of biological and synthetic targeted therapies (bDMARD/tsDMARD respectively) and the “treat to target” strategy [1,2,3]. However, despite recent advances, the therapy of RA is still challenging, since approximately one third of the patients lack a satisfactory response to treatment, associated with decreased quality of life due to persistent pain, fatigue, and impairment of physical and mental activity [4,5].

Several genetic, epigenetic, and environmental risk factors, such as infections, smoking, diet, specific microbiome, etc., can contribute to the abnormal activation of the immune system in RA. Different types of adaptive and innate immune cells, such as B and T cells, neutrophils, mast cells, macrophages, and dendritic cells, as well as synovial fibroblasts involved in the pathogenesis, produce a wide range of inflammatory cytokines (e.g., tumor necrosis factor-α (TNF-α), interleukin (IL)-1β, IL-6, IL-8, IL-12/IL-23, IL-17, IL-18, IL-32, and interferon-γ (IFN-γ)) that are responsible for further enhancing inflammatory processes [1].

Janus kinase signaling (JAK1-, JAK2-, JAK3- and tyrosine kinase 2/TYK2) linked to the signal transducer and activator of transcription (STAT) pathway is directly or indirectly associated with most of these cytokines and therefore plays a key role in the inflammatory mechanisms in RA. JAKs are non-receptor tyrosine kinases activated by the binding of certain ligands to their membrane bound receptor. Once activated, JAKs phosphorylate the associated receptor forming a docking site for the members of the STAT protein family. After docking, STAT proteins also get phosphorylated by JAKs leading to their dissociation and dimerization. The activated STAT dimers then translocate to the nucleus and take part in the regulation of transcription [6]. This process can be selectively blocked by small molecule JAK inhibitors [7]. In addition, they have been described to directly inhibit pain processing [8], since the receptors of many inflammatory cytokines regulating through the JAK-STAT pathway are expressed on neuronal and glial cells [8], but these neuronal mechanisms are currently poorly understood.

Currently, JAK inhibitors such as tofacitinib, baricitinib, upadacitinib, filgotinib, and peficitinib play a predominant role in RA treatment, and other more recent ones like decernotinib, ritlecitinib, and itacitinib are in phase II clinical trials. JAK inhibitors are effective and safe, therefore, they are recommended as first-choice treatment by the European Alliance of Associations for Rheumatology (EULAR) [3] and the American College of Rheumatology (ACR) [9,10] in patients inadequately responding to methotrexate (MTX), equivalent to bDMARDs [3,9,10]. Moreover, several studies also examined JAK inhibitor monotherapy in both MTX naïve and resistant populations.

Here, we provide a comprehensive summary of the latest evidence with the efficacy and safety of the currently used and investigated JAK inhibitors, especially focusing on pain and PROs.

## 2. Results

### 2.1. Description of the Studies

A systematic search resulted in 3020 hits; two additional studies were identified through searching the citations of eligible works. After the removal of duplicates, 1703 articles remained, and 1383 further records were excluded based on the title and abstract screening process. The full texts of the remaining 320 articles were assessed for inclusion, and finally, 55 individual randomized controlled trials (RCT) from 88 publications were identified to be eligible for the qualitative synthesis from which 50 were used for the quantitative evaluation (Figure 1).

All 50 studies summarizing the data of 24,135 active RA patients were RCTs comparing JAK inhibitors to either placebo or other therapeutic agents. In 40 studies, patients showed an inadequate response to different types of csDMARD and bDMARD therapies [11,12,13,14,15,16,17,18,19,20,21,22,23,24,25,26,27,28,29,30,31,32,33,34,35,36,37,38,39,40,41,42,43,44,45,46,47,48,49,50], in 5 articles, patients were naïve to csDMARDs [51,52,53,54,55], from which 4 to MTX [51,52,53,54], 1 study did not give restrictions regarding previous medication [56], and 4 studies did not include information about the treatment history [57,58,59]. All JAK inhibitors approved for RA patients were included: tofacitinib appears in 16 [11,12,13,14,15,16,17,18,19,20,21,22,23,51,54,57], baricitinib in 8 [24,25,26,27,28,29,30,55,60], upadacitinib in 9 [31,32,33,34,35,36,37,38,52], filgotinib in 7 [39,40,41,42,53,58], peficitinib in 5 [43,44,45,46,56], decernotinib in 3 articles [47,48,49], and both ritlecitinib [50] and itacitinib [59] were investigated in 1 study. In 14 studies, patients received JAK monotherapy [15,17,20,23,31,41,43,49,51,52,53,54,55,56], whereas concomitant therapy was applied in 36 studies: one study allowed patients with any type of DMARDs [47], one only nonbiologic DMARDs [13], 10 only csDMARDs [24,25,30,33,34,37,38,40,59,60], 23 specifically MTX [11,12,14,16,18,19,21,22,27,28,29,32,34,35,39,42,44,45,46,47,50,58], and in one study [57] data could not be obtained. Short-term (<6 months) and long-term (>6 months) effects of JAK inhibitors were analyzed separately, however, for most of the outcomes, sufficient data was only present to assess short-term therapy. Detailed information of the included RCTs is collected in Table 1 and Appendix A, whereas a summary of the grading of the quality of evidence is reported in Table 2.

### 2.2. JAK Inhibitors vs. Placebo

A total of 43 articles compared different types of JAK inhibitors to placebo [11,12,13,14,15,17,19,20,21,22,23,24,25,27,28,29,30,32,33,34,35,36,38,39,40,41,42,43,44,45,46,47,48,49,50,51,53,55,56,58,59,60]; meta-analysis was possible for 36 different outcomes. Efficacy was monitored via disease activity, inflammatory parameters, and patient-reported questionnaires; in addition, safety information was also collected. ACR 20/50/70 response rates were assessed, and disease activity was measured by the proportion of patients reaching remission as defined by the Disease Activity Score 28 using C-reactive protein (DAS28-CRP), the DAS28 using erythrocyte sedimentation rate (DAS28-ESR), the Simplified Disease Activity Index (SDAI), and the Clinical Disease Activity Index (CDAI) and was also characterized by the mean change from baseline values in the different disease activity parameters (Figure 2). ACR response rates were considerably higher (ACR20: odds ratio (OR) 3.31, 95% confidence interval (CI) 2.85 to 3.86, *p* < 0.001), furthermore, treatment with JAK inhibitors also showed a statistically significant advantage in all disease activity parameters over placebo within 6 months (Figure 1). CRP (weighted mean difference (WMD): −8.57 mg/L, 95% CI −10.14 to −6.99, *p* < 0.001) and ESR (WMD −14.17 mm/h 95% CI −19.28 to −9.07, *p* < 0.001) also significantly decreased in the JAK-treated group. The analgesic effect of JAK inhibitors was also demonstrated (WMD of pain measured on visual analogue scale (VAS) −15.29, 95% CI −17.34 to −13.24, *p* < 0.001); most patient-reported outcomes also significantly improved compared to the placebo group. In 23 of the examined efficacy outcomes, the optimal information size was reached according to the trial sequential analysis (TSA). Although the proportion of adverse events was higher among patients treated with JAK inhibitors (OR 1.20, 95% CI 1.07 to 1.34, *p* = 0.002), the number of serious side effects did not differ significantly (OR 0.95, 95% CI, 0.75 to 1.21, *p* = 0.687) between the two arms. It is noteworthy that the addition of further studies could influence these results based on the TSAs. Discontinuation of the medication also occurred similarly (OR 1.03, 95% CI 0.81 to 1.31, *p* = 0.782). No significant difference was observed in the proportion of patients who died during the study (OR 0.81, 95% CI 0.40 to 1.63, *p* = 0.559). These results are summarized in Figure 2, whereas individual forest plots and TSAs can be found in the Appendix A.

Green diamonds represent the overall effect, whereas the outer lines are the confidence intervals. ACR denotes American College of Rheumatology; CDAI, clinical disease activity index; CRP, C-reactive protein; DAS28-CRP, disease activity score 28 using C-reactive protein; DAS28-ESR, disease activity score 28 using erythrocyte sedimentation rate; ESR, erythrocyte sedimentation rate; EQ-5D UK, EuroQol 5 dimensions questionnaire—UK scoring algorithm; EQ-5D US, EuroQol 5 dimensions questionnaire—US scoring algorithm, EQ-5d (VAS), EQ-5D measure on visual analogue scale; FACIT-F, functional assessment of chronic illness therapy—fatigue; HAQ-DI, health assessment questionnaire—disability index; JAKi, JAK inhibitor; MOS-sleep, medical outcomes study sleep scale; PGA; MJS, morning joint stiffness; physician’s global assessment of disease activity; PtGA, patient’s global assessment of disease activity; SDAI, simple disease activity index; SF-36 MCS, 36-item short form survey—mental component score, SF-36 PCS, SF-36 physical component score; WPAI A, WPAI—absenteeism; WPAI AI, work productivity and activity impairment questionnaire—activity impairment; WPAI—P, WPAI presenteeism; WPAI—OWI, WPAI overall work impairment.

### 2.3. JAK Inhibitors vs. MTX

Six RCTs focused on JAK monotherapy versus MTX [31,51,52,53,54,55], and five of them included an MTX-naïve population [51,52,53,54,55]. A total of 12 different outcomes were analyzed, and the optimal information size was reached for only 4 parameters (Figure 3). ACR response rates were significantly higher among patients treated with JAK inhibitors (ACR20: OR 2.33, 95% CI 1.80 to 3.03, *p* < 0.001), and a higher proportion of patients were in remission as defined by the CDAI (OR 3.63 95 % CI 1.33 to 9.90, *p* = 0.012) and DAS-28-CRP (OR 3.05, 95% CI 1.79 to 5.18, *p* = 0.002) within 6 months. However, patients reaching remission according to the SDAI (OR 3.21, 95% CI 0.99 to 10.4, *p* = 0.052) and DAS-28-ESR (OR 2.62, 95% CI 0.85 to 8.10, *p* = 0.095) did not differ significantly between the two treatment arms. Quality of life measured using different PROs showed significantly greater improvement compared to MTX. A similar proportion of patients experienced serious side effects in both groups (OR 1.30, 95% CI 0.75 to 2.25, *p* = 0.356). Patients were already exposed to MTX therapy in one of the studies, which could be responsible for the considerable heterogeneity appearing in those outcomes, where it is involved [31]. Secondary analyses were conducted for these outcomes (ACR 50, 70 and DAS28-ESR), and the improvement of heterogeneity could be observed in all cases. These results are summarized in Figure 3, whereas individual forest plots, results of the secondary analyses, and TSAs can be found in the Appendix A.

Green diamonds represent the overall effect, whereas the outer lines are the confidence intervals. ACR denotes American College of Rheumatology; CDAI, clinical disease activity index; DAS28-CRP, disease activity score 28 using C-reactive protein; DAS28-ESR, disease activity score 28 using erythrocyte sedimentation rate; HAQ-DI, health assessment questionnaire—disability index; FACIT-F, functional assessment of chronic illness therapy—fatigue; JAKi, JAK inhibitor; SDAI, simple disease activity index; and SF-36 PCS, 36-item short form survey—physical component score.

### 2.4. JAK Inhibitors vs. bDMARDs

Six studies compared JAK inhibitors to the anti-TNF-α monoclonal antibody adalimumab [12,16,17,24,32,39], 2 to the soluble TNF receptor etanercept [46,57], and 2 to non-TNF inhibitor bDMARDs, tocilizumab, and abatacept [18,37]. Meta-analyses were performed in the case of 21 outcomes; the optimal information size was reached in all except for one (number of deaths). JAK inhibitors proved to be statistically superior to bDMARDs in response rate according to the ACR criteria (ACR20: OR, 1.30, 95% CI, 1.15 to 1.48, *p* = *p* < 0.001) within 6 months. The proportion of patients reaching remission defined by the DAS-28-CRP (OR 1.95, 95% CI 1.27, 3.00) were significantly higher compared to bDMARDs, but based on the remission threshold of the DAS-28-ESR (OR 2.62, 95% CI 0.85 to 8.10, *p* = 0.288), CDAI (OR 1.51, 95% CI 0.99 to 2.30, *p* = 0.055), and SDAI (OR 1.51, 95% CI 0.99 to 2.30, *p* = 0.057), JAK inhibitors and bDMARDs were similarly efficient in decreasing disease activity in the first 6 months. However, in these cases, TSAs revealed that additional studies might be able to alter these results. Regarding inflammation, JAK inhibitors also reduced CRP at a greater extent (WMD −3.94 mg/L, 95% CI −5.35 to −2.52, *p* < 0.001). Their ability to decrease pain (WMD of pain measured on VAS −4.35, 95% CI −6.47 to −2.23, *p* < 0.001) and tender joint (OR −1.46, 95% CI −2.18 to −0.74, *p* < 0.001) and swollen joint count (OR −0.60, 95% CI 1.10 to −0.10, *p* = 0.019) proved to be significantly greater than in the case of biologicals. Concerning other PROs, JAK inhibitors reached significantly better results in swollen joint count, tender joint count, and 2 of the questionnaires regarding fatigue and general health status. However, other surveys describing life-quality measures did not differ significantly. No significant differences were noted in serious side effects (OR 1.56, 95% CI 0.82 to 2.98, *p* = 0.174) and the number of deaths (OR 0.76 95% CI 0.09 to 6.53, *p* = 0.801). These data are summarized in Figure 4, whereas individual forest plots and TSAs are presented in the Appendix A.

Green diamonds represent the overall effect, whereas the outer lines are the confidence intervals. ACR denotes American College of Rheumatology; CDAI, clinical disease activity index; CRP, C-reactive protein; DAS28-CRP, disease activity score 28 using C-reactive protein; DAS28-ESR, disease activity score 28 using erythrocyte sedimentation rate; FACIT-F, functional assessment of chronic illness therapy—fatigue; HAQ-DI, health assessment questionnaire—disability index; JAKi, JAK inhibitor; MJS, morning joint stiffness; PGA; physician’s global assessment of disease activity; PtGA, patient’s global assessment of disease activity; SDAI, simple disease activity index; SF-36 MCS, 36-item short form survey—mental component score, SF-36 PCS, SF-36 physical component score; and WPAI—OWI, work productivity and activity impairment questionnaire—overall work impairment.

## 3. Discussion

JAK inhibitors are effective medications in RA [3]. Currently, tofacitinib, baricitinib, and upadacitinib are widely used worldwide, filgotinib has also been approved in Europe and Japan, and peficitinib in Japan. Furthermore, several other JAK inhibitors, such as decernotinib, ritlecitinib, and itacitinib are being studied in clinical trials [47,48,49,50,59] to evaluate their efficacy and safety in RA. This is the first meta-analysis, which offers a comprehensive overview of the effects of JAK inhibitors currently being used or under clinical development with 36 integrative outcomes compared to placebo and other DMARDs. Our results provide one of the first pieces of evidence for the ability of JAK inhibitors to improve PROs including pain in addition to the conventional inflammatory parameters.

RA highly effects different aspects of life quality, and among all of the symptoms, pain is the most dominant for patients [91]. Pain originates from a broad range of sensitization mechanisms associated with systemic inflammatory processes, neuroinflammation, structural joint damage, and comorbidities [92] (Appendix A). Several cytokines (e.g., TNF-α, IL-1β, IL-6, IL-17) which regulate the inflammatory processes can also directly contribute to the induction and maintenance of pain [8,93,94]. Cytokines stimulate their receptors expressed on the primary sensory nerve endings innervating the synovia and joint capsule, their cell bodies in the dorsal root ganglia [93,94], and secondary nociceptive neurons in the spinal cord [95], leading to pain sensitization and hyperalgesia. Astrocytes and microglial cells also produce substantial amount of these cytokines [96,97], which trigger consequent complex neuro-immune interactions, leading to accelerated pain, anxiety, and depression [98]. The JAK-STAT cascade is involved in inflammatory cytokine signaling in the nervous system indicating its direct role in pain sensitization. In contrast to a single cytokine blockade by bDMARDs, JAK inhibitors inhibit the effect of multiple cytokines [8,99]. The direct analgesic effect of JAK inhibitors was suggested by RCTs [24,32,65,68], e.g., RA-BEAM [24], showing that baricitinib is superior to adalimumab in alleviating pain, with a similar anti-inflammatory effect. This observation is supported by the results of our extensive meta-analysis, demonstrating that JAK inhibitors showed a significantly greater pain-relieving effect compared to bDMARDs. In addition, the CRP values indicating the intensity of the inflammatory reaction were also lower in the JAK inhibitor group. Previously, no other meta-analysis examined the effect of JAK inhibitors on PROs, however, some studies included outcomes regarding life quality. One analysis included an improvement in the HAQ-DI and found JAK inhibitors overall as effective as bDMARDs, except for tofacitinib 10 mg showing significant advantage compared to adalimumab and baricitinib 2 mg performing slightly worse than bDMARDs [100]. In our analysis, the different JAK inhibitors were pooled into one group and significant differences between JAK inhibitors versus bDMARDs were not observed either. The SF-36 questionnaire exploring mental health was also examined in some studies. One of them found JAK inhibitors slightly more effective than bDMARDs [101], however, another study revealed no significant difference, similarly to our results [102].

Importantly, the proportion of patients reaching remission assessed using the DAS28-CRP was significantly higher in the JAK group than in the case of bDMARDs, which is consistent with the findings of a previous meta-analysis demonstrating the superiority of upadacitinib compared to adalimumab [100]. However, no statistically significant differences were found in remission rates according to the DAS28-ESR, SDAI, and CDAI. This difference might be explained by the observation that the DAS28-CRP can underestimate the disease activity [103].

Our analysis also demonstrated the superiority of JAK inhibitors compared to bDMARDs in several other efficacy outcomes, e.g., ACR20/50/70, in line with previous meta-analyses also describing the advantages of tofacitinib and baricitinib over adalimumab regarding efficacy outcomes [104,105,106]. Although this difference was statistically significant in most outcomes, the clinical relevance of the individual results might seem uncertain. JAK inhibitors have other advantages compared to bDMARDs as well. bDMARDs tend to lose their efficacy with time, which might partially be explained by the formation of anti-drug antibodies [107]. In addition, JAK inhibitors are small molecules appropriate for oral administration, which makes patients’ compliance better. Therefore, our results, together with other advantageous features of JAK inhibitors, clearly suggest their benefits over bDMARDs.

MTX is usually the initial treatment option in RA [3], but in many patients therapy with MTX alone is not sufficient to control the disease and the addition of a b/tsDMARD is often needed [108]. However, several studies also examine the difference between JAK inhibitor monotherapy and MTX in an MTX-naïve population, suggesting the possibility of an early administered JAK inhibitor monotherapy. Our results confirm and extend the findings of a previous network meta-analysis, demonstrating the significant advantage of different JAK inhibitors in ACR 50 and 70 response compared to MTX alone [109]. We found the superiority of JAK inhibitor monotherapy compared to MTX in all investigated outcomes, except for the number of patients in remission defined using the DAS-28-ESR and SDAI. On the other hand, significantly more patients reached remission thresholds defined using the DAS-28-CRP and CDAI as well as achieved ACR20/50/70 response rates in the JAK inhibitor group, clearly supporting the overall efficiency of JAK inhibitor monotherapy compared to MTX, especially in contrast to bDMARDs, which seem more advantageous in combination therapy [110].

Our analysis has some limitations. Firstly, most of the included trials did not exceed six months or switched therapy beforehand, therefore, these results should be interpreted only for the short-term effect of JAK inhibitors, since the long-term effect could not be observed effectively. The completion of the currently ongoing follow-up studies as well as newly conducted analyses would be necessary to draw a clearer conclusion. Secondly, considerable statistical heterogeneity appeared in some of our analyses, which can be explained by evaluating JAK inhibitors together. On the other hand, although the demographic characteristics of the included patients in the RCTs, the study designs, or the different types of the permitted concomitant therapy were in general comparable, some smaller differences could be observed, which could also be responsible for the heterogeneity. However, the chi-square (χ2) test is capable of detecting even a small amount of heterogeneity in the case of a meta-analysis including a large amount of RCTs, just as in our analysis, without necessarily indicating clinically relevant heterogeneity [111]. Furthermore, secondary analyses also helped to find the cause of heterogeneity in some comparisons. Finally, most of the RCTs included patients with moderate-to-severe RA. Although this is in line with clinical practice, where JAK inhibitors are administered when remission or low disease activity cannot be reached [3], our findings should be interpreted with caution.

Our study has several strengths, such as the profound, integrative, and up-to-date analysis with complex, multiple comparison, as well as the inclusion of additional drugs and outcomes. We mainly focus on pain and other PROs, which is often neglected in most of the studies, however, it is one of the most important symptoms for the patients. Due to the inclusion of numerous RCTs providing many participants, our results have adequate statistical power. In addition, the optimal information size was reached in most of our outcomes, suggesting that the involvement of additional trials is unlikely to change the result of our analysis.

## 4. Materials and Methods

A systematic literature search was conducted to obtain a comprehensive view and evidence on the efficacy of JAK inhibitors in RA patients, mainly focusing on their anti-inflammatory and analgesic actions. This study was carried out in accordance with the Preferred Reporting Items for Systematic Reviews and Meta-Analyses (PRISMA) statement [112] and the recommendations included in the Cochrane Handbook for Systematic Reviews of Interventions [111]. The protocol for this study was registered with PROSPERO (registration number: CRD42021222899 on 8 January 2021). No deviations were made from the preregistered protocol.

### 4.1. Search Strategy

Appropriate studies were identified via the systematic search of MEDLINE (via PubMed), Embase, CENTRAL, Web of Science, and Scopus electronic scientific databases conducted on 26 October 2020, with the following search key: “(rheumatoid arthritis) AND (JAK OR (Janus kinase) OR tofacitinib OR CP690550 OR LY3009104 OR baricitinib OR upadacitinib OR filgotinib OR Olumiant OR Xeljanz OR incb028050 OR “ABT-494” OR Rinvoq OR peficitinib OR ASP015K OR decernotinib OR “VX-509” OR tasocitinib OR GLPG0634 OR “GS-6034” OR ritlecitinib OR “pf 6651600”) AND random *”. No filters were applied during the search. The reference list of the identified articles and previous reviews were also checked manually to find additional studies.

### 4.2. Selection Criteria

Articles were selected on the basis of our research question following the population-intervention-comparator-outcome (PICO) format. Patients with active RA above the age of 18 years were chosen as the population, and different types of JAK inhibitors were investigated in comparison with (1) placebo, (2) bDMARDs in patients with inadequate response to conventional synthetic DMARDs (csDMARD) or previous bDMARDs, and (3) with MTX as monotherapy in MTX naïve and resistant populations. Our main goal was to give a comprehensive picture of the efficacy and safety of JAK inhibitors; therefore, data were collected for all outcomes where sufficient information was provided (Appendix A). Pain and other PROs were our main focus topics; therefore, pain, patient’s global assessment of disease activity (PtGA) and physician’s global assessment of disease activity (PGA) were chosen as primary outcomes. However, different inflammatory and disease activity markers were also evaluated in our analysis. Studies were limited to RCTs; no other restrictions were applied. Eligible conference abstracts of RCTs were also included.

### 4.3. Selection Process and Data Extraction

The selection process was performed using EndNote X9 software (Clarivate Analytics, Philadelphia, PA, USA). After duplicate removal, records were screened via the title, abstract, and full text according to a previously discussed set of rules.

Data were extracted into standardized data collection sheets created with Excel (Microsoft, Office 365, Redmond, WA, USA), especially for this purpose. Collected information consisted of data regarding the article (author, publication date, DOI, language, study design and duration, number and region of the participating centers), regarding the population (total, randomized and completed number of patients, age, gender, inclusion and exclusion criteria, concomitant therapy in each treatment arm), and regarding intervention and comparators (specific drugs or placebo, doses, route and frequency of administration, treatment duration, number of patients). Information for altogether 36 outcomes (Appendix A) were extracted. The outcomes were examined at different time points, ranging from 4 weeks to 2 years. Although the vast majority of the results were within 6 months if sufficient data were provided, in order to appropriately compare the results, different meta-analyses were conducted for studies within and over 6 months, defined as short-term and long-term examinations. Duplicate reports (e.g., conference abstracts, secondary analyses, etc.) of an eligible study population were also processed if providing additional data, and these records were linked to the original publications.

The selection and data extraction process were performed by three authors (AA, FK, and LT). Any discrepancies were first discussed by the investigators. If consensus could not be reached, disagreements were resolved with the help of an expert investigator (MFJ).

### 4.4. Statistical Analysis

If at least three RCTs provided sufficient data for an outcome, a meta-analysis was performed, and the results were displayed on forest plots. A random effects model was applied, and depending on the type of outcome, for dichotomous variables pooled ORs and for continuous outcomes pooled WMDs with their 95% CI were calculated to investigate the differences between the compared treatment arms. In all instances raw data were used: in the case of binary data, number of event and non-event, in the case of continuous data, mean and standard deviation. Statistical heterogeneity was assessed using the χ2 test and I-squared (I2) statistic, where *p* < 0.1 was considered as statistically significant and an I2 value of 0–40%, 30–60%, 50–90%, and 75–100% indicated not important, moderate, substantial, and considerable heterogeneity based on the Cochrane Handbook’s recommendation. If there was any noticeable heterogeneity, to reveal the cause, secondary analyses were performed.

To detect potential publication bias, Egger’s test was performed if there were at least 10 studies eligible; in all other cases visual inspection of funnel plots (Appendix A) was applied. All calculations were performed with the Stata Statistical Software (StataCorp 2019. Stata Statistical Software: Release 16. StataCorp.LLC, College Station, Texas, USA). In order to reduce the risk of type I and type II errors and determine the required information size, TSAs (TSA Application, Copenhagen Trial Unit, Center for Clinical Intervention Research, Copenhagen, Denmark) were performed applying a two-sided alpha of 0.05 and a beta of 0.20 [113].

### 4.5. Assessment of Risk of Bias and the Quality/Certainty of Evidence

Two authors (AA and LT) independently assessed the potential risk of bias in the included studies, with the use of the revised risk of bias 2 (ROB 2) tool [114] based on the recommendations of Cochrane Collaboration (Appendix A). The certainty of evidence was evaluated using the Grading of Recommendations Assessment, Development and Evaluation (GRADE) [115] system (Table 2). Disagreements were discussed first and, in the case of further discrepancies, were resolved by an independent third researcher (MFJ).

## 5. Conclusions

Pain is one of the most crucial symptoms for RA patients leading to impaired quality of life, motility, and working capability. This comprehensive and detailed systematic review and meta-analysis provided evidence for the efficacy and safety of JAK inhibitors with specific emphasis on pain and other PROs. They proved to be highly effective in all investigated outcomes in comparison with placebo, without raising any safety concerns. In contrast to bDMARDs, JAK inhibitors showed significant advantages not just in improving quality of life, but also reducing inflammation in patients with inadequate response to at least one of either csDMARD or bDMARD. Furthermore, we also proved their significant efficacy in reducing disease activity over MTX monotherapy. Future studies investigating the long-term differences of JAK inhibitors in comparison to other medications as well as head-to-head trials of the different JAK inhibitors would be important to further evaluate the different aspects of their effectiveness.

## Figures and Tables

**Figure 1 ijms-23-01246-f001:**
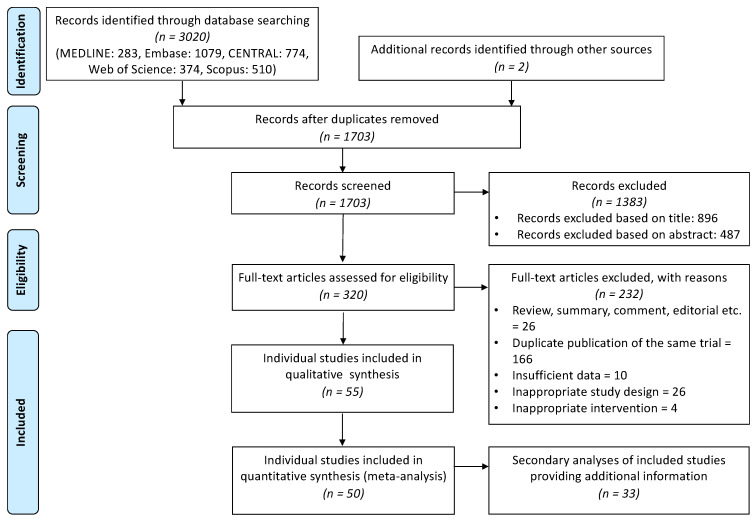
Prisma flowchart demonstrating the results of selection process.

**Figure 2 ijms-23-01246-f002:**
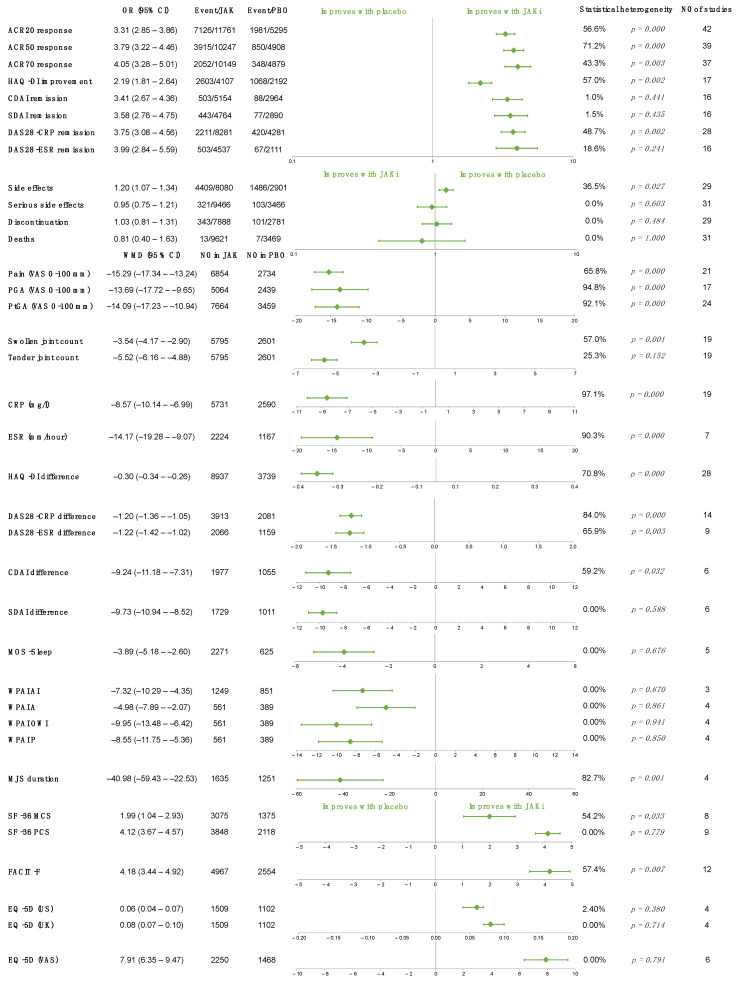
Overall results of each outcome evaluated in JAK inhibitor versus placebo comparison.

**Figure 3 ijms-23-01246-f003:**
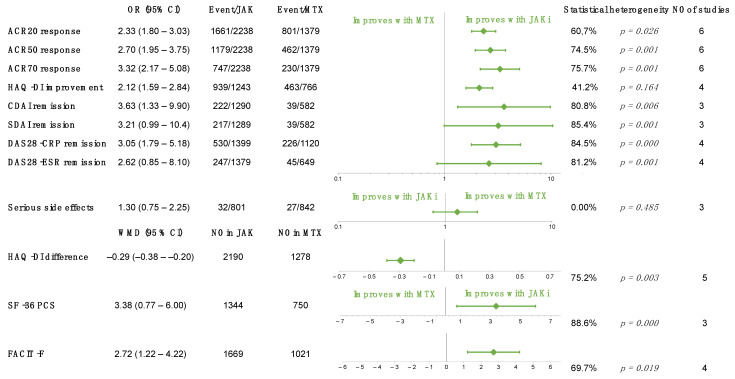
Overall results of each outcome evaluated in JAK inhibitor versus MTX comparison.

**Figure 4 ijms-23-01246-f004:**
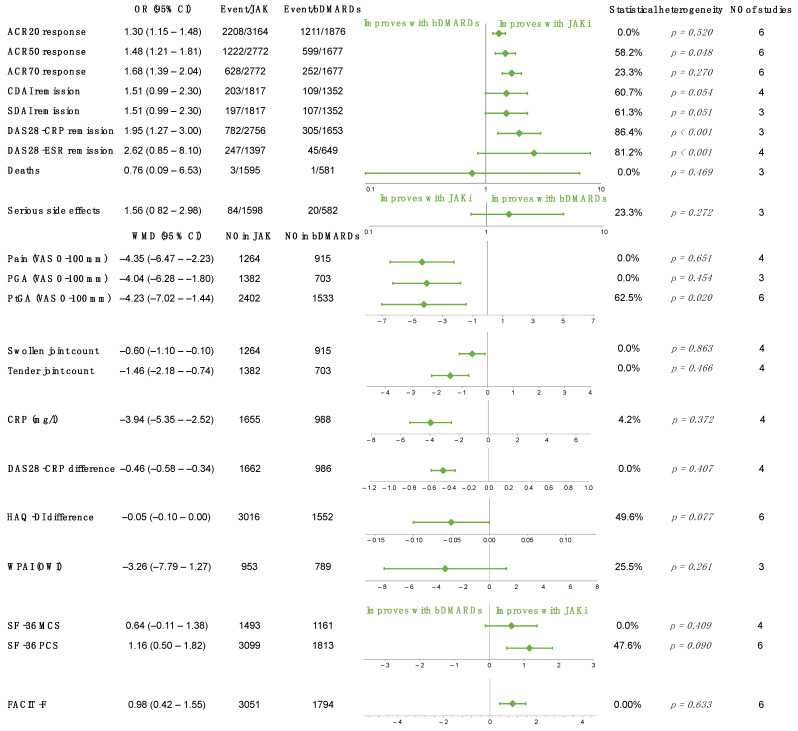
Overall results of each outcome evaluated in JAK inhibitor versus bDMARDs comparison.

**Table 1 ijms-23-01246-t001:** Characteristics of the included studies.

Study	Year	Population	Total No of Patients	Intervention	Comparator		Concomitant Medication
					PBO	Active Comparator	
**NCT00902486** **[30]**	2010	inadequate response to DMARDs	124	baricitinib	+	–	csDMARD
**NCT01185353** **[28,61,62]**	2015	inadequate response to MTX	301	baricitinib	+	–	MTX
**NCT02265705** **[27,63]**	2020	inadequate response to MTX	290	baricitinib	+	–	MTX
**NCT01721044** **[25,64]**	2016	inadequate response to TNFi	527	baricitinib	+	–	csDMARD
**NCT01710358** **[24,65]**	2017	inadequate response to MTX	1305	baricitinib	+	adalimumab	csDMARD
**NCT01711359** **[55,66,67]**	2017	csDMARD naive	584	baricitinib	+	MTX	–
**NCT01721057** **[26,60,68]**	2017	inadequate response to csDMARDs	684	baricitinib	+	–	csDMARDs
**NCT01469013** **[29]**	2016	inadequate response to MTX	145	baricitinib	+	–	MTX
**NCT01052194** **[49,69]**	2015	inadequate response to MTX	204	decernotinib	+	–	–
**NCT2011–004419–22** **[47,70]**	2016	inadequate response to MTX	358	decernotinib	+	–	MTX
**NCT01754935** **[48]**	2016	inadequate response to DMARDs	43	decernotinib	+	–	DMARDs
**NCT01888874** **[42]**	2016	inadequate response to MTX	594	filgotinib	+	–	MTX
**NCT01894516** **[41]**	2017	inadequate response to MTX	283	filgotinib	+	–	–
**NCT02889796** **[39]**	2019	inadequate response to MTX	1755	filgotinib	+	adalimumab	MTX
**NCT02873936** **[40]**	2019	inadequate response or intolerance to bDMARDs	448	filgotinib	+	–	csDMARD
**NCT02886728** **[53]**	2019	MTX naive	1249	filgotinib	+	MTX	–
**NCT01384422** **[58]**	2017	n.a.	36	filgotinib	+	–	MTX
**NCT01668641** **[58]**	2017	n.a.	91	filgotinib	+	–	MTX
**NCT01626573** **[59]**	2013	n.a.	60	itacitinib	+	–	csDMARDs
**NCT01565655** **[43,71]**	2017	inadequate response to csDMARDs	289	peficitinib	+	–	–
**NCT01554696** **[44,72]**	2017	inadequate response to MTX	379	peficitinib	+	–	MTX
**NCT02308163** **[46]**	2019	inadequate response to csDMARDs	507	peficitinib	+	etanercept	MTX
**NCT02305849** **[45]**	2019	inadequate response to MTX	519	peficitinib	+	–	MTX
**NCT01649999** **[56]**	2016	no restrictions	281	peficitinib	+	–	–
**NCT02969044** **[50]**	2020	inadequate response to MTX	70	ritlecitinib	+	–	MTX
**NCT00976599** **[19]**	2015	inadequate response to MTX	29	tofacitinib	+	–	MTX
**NCT01164579** **[51]**	2016	MTX naive	109	tofacitinib	+	MTX	–
**NCT00550446** **[17]**	2012	inadequate response to DMARDs	384	tofacitinib	+	adalimumab	–
**NCT00147498** **[20,73]**	2009	inadequate response to DMARDs	264	tofacitinib	+	–	–
**NCT00413660** **[21]**	2012	inadequate response to DMARDs	507	tofacitinib	+	–	MTX
**Menshikova 2018,** **[57]**	2018	n.a.	30	tofacitinib	–	etanercept	n.a.
**NCT02157012** **[18]**	2018	inadequate response to DMARDs	50	tofacitinib	–	tocilizumab, abatacept	MTX
**NCT00847613** **[11,74]**	2013	inadequate response to MTX	797	tofacitinib	+	–	MTX
**NCT00814307** **[17,75]**	2012	inadequate response to DMARDs	610	tofacitinib	+	–	–
**NCT00853385** **[12,76,77]**	2012	inadequate response to MTX	717	tofacitinib	+	adalimumab	MTX
**NCT01039688** **[54,78,79]**	2014	MTX naive	956	tofacitinib	–	MTX	–
**NCT00960440** **[14,80]**	2013	inadequate response to TNFi	399	tofacitinib	+	–	MTX
**NCT02187055** **[16,81,82]**	2017	inadequate response to MTX	1146	tofacitinib	–	adalimumab	MTX
**NCT00856544** **[13,83]**	2013	inadequate response to DMARDs	792	tofacitinib	+	–	nonbiologic DMARDs
**NCT00603512** **[22]**	2011	inadequate response to MTX	136	tofacitinib	+	–	MTX
**NCT00687193** **[23]**	2015	inadequate response to at least one synthetic or bDMARD	317	tofacitinib	+	–	–
**NCT01960855** **[35]**	2016	inadequate response to TNFi	276	upadacitinib	+	–	MTX
**NCT02066389** **[34]**	2016	inadequate response to MTX	299	upadacitinib	+	–	MTX
**NCT02706847** **[36,84,85]**	2018	inadequate response to bDMARDs	499	upadacitinib	+	–	csDMARD
**NCT03086343** **[37]**	2020	inadequate response to bDMARDs	612	upadacitinib	–	abatacept	csDMARD
**NCT02629159** **[32,86]**	2019	inadequate response to MTX	1629	upadacitinib	+	adalimumab	MTX
**NCT02706873** **[52,87,88]**	2020	MTX naive	945	upadacitinib	–	MTX	–
**NCT02706951** **[31,89]**	2019	inadequate response to MTX	648	upadacitinib	–	MTX	–
**NCT02675426** **[33,90]**	2018	inadequate response to csDMARDs	661	upadacitinib	+	–	csDMARD
**NCT02720523** **[38]**	2020	inadequate response to csDMARDs	197	upadacitinib	+	–	csDMARD

All included studies are RCTs. DMARD denotes disease modifying antirheumatic drugs; csDMARD, conventional synthetic DMARD; bDMARD, biological DMARD; TNFi, tumor necrosis factor inhibitor; MTX, methotrexate.

**Table 2 ijms-23-01246-t002:** Summary of findings and certainty of evidence.

Outcomes	No. of Participants (Studies) Follow Up	Certainty of the Evidence (GRADE)	Relative Effect (95% CI)	Anticipated Absolute Effects	
				Risk with Placebo	Risk Difference with JAK Inihbitors
JAK inhibitors compared to placebo (<6 months)					
Pain assessed on VAS Scale from: 0 to 100	9588 (21 RCTs)	⊕⊕⊕◯ MODERATE ^a^^,^^b^	–	The mean pain was 0 mm	MD 15.29 mm lower (17.34 lower to 13.24 lower)
Number of patients reaching remission according to DAS28–ESR	6648 (16 RCTs)	⊕⊕⊕⊕ HIGH ^b^	OR 3.99 (2.84 to 5.59)	32 per 1000	84 more per 1000 (53 more to 123 more)
Number of patients reaching remission according to DAS28–CRP	12,562 (28 RCTs)	⊕⊕⊕◯ MODERATE ^a^^,^^b^	OR 3.75 (3.08 to 4.56)	98 per 1000	192 more per 1000 (153 more to 233 more)
Number of patients reaching 20% improvement according to ACR criteria (ACR20)	17,056 (42 RCTs)	⊕⊕⊕◯ MODERATE ^a^^,^^b^	OR 3.31 (2.85 to 3.86)	374 per 1000	290 more per 1000 (256 more to 324 more)
Mortality	13,090 (31 RCTs)	⊕⊕◯◯ LOW ^b^	OR 0.81 (0.40 to 1.63)	2 per 1000	0 fewer per 1000 (1 fewer to 1 more)
Number of patients with serious side effects	12,932 (31 RCTs)	⊕⊕⊕◯ MODERATE ^a^	OR 0.95 (0.75 to 1.21)	30 per 1000	1 fewer per 1000 (7 fewer to 6 more)
Change in CRP assessed with: mg/L	8321 (19 RCTs)	⊕⊕◯◯ LOW ^a^^,^^b^	–	The mean change in C–reactive protein was 0 mg/L	MD 8.57 mg/L lower (10.14 lower to 6.99 lower)
JAK inhibitors compared to bDMARDs (<6 months)					
Pain assessed on VAS Scale from: 0 to 100	2179 (4 RCTs)	⊕⊕⊕◯ MODERATE ^a^	–	The mean pain was 0 mm	MD 4.35 mm lower (6.47 lower to 2.23 lower)
Number of patients reaching remission according to DAS28–ESR	2046 (4 RCTs)	⊕◯◯◯ VERY LOW ^a^^,^^b^	OR 2.62 (0.85 to 8.10)	69 per 1000	94 more per 1000 (10 fewer to 307 more)
Number of patients reaching remission according to DAS28–CRP	4409 (3 RCTs)	⊕⊕◯◯ LOW ^a^^,^^b^	OR 1.95 (1.27 to 3.00)	185 per 1000	122 more per 1000 (39 more to 220 more)
Number of patients reaching 20% improvement according to ACR criteria (ACR20)	5040 (6 RCTs)	⊕⊕⊕◯ MODERATE ^a^	OR 1.30 (1.15 to 1.48)	646 per 1000	58 more per 1000 (31 more to 84 more)
Mortality	2176 (3 RCTs)	⊕⊕◯◯ LOW ^a^	OR 0.76 (0.09 to 6.53)	2 per 1000	0 fewer per 1000 (2 fewer to 9 more)
Number of patients with serious side effects	2180 (3 RCTs)	⊕⊕◯◯ LOW ^a^	RR 1.56 (0.82 to 2.98)	34 per 1000	19 more per 1000 (6 fewer to 68 more)
Change in CRP assessed with: mg/L	2643 (4 RCTs)	⊕⊕◯◯ LOW ^a^	–	The mean change in C–reactive proteint was 0 mg/L	MD 3.94 mg/: lower (5.35 lower to 2.52 lower)
JAK inhibitors compared to MTX (<6 months)					
Pain—not measured	–	–	–	–	–
Number of patients reaching remission according to DAS28–ESR	2028 (4 RCTs)	⊕◯◯◯ VERY LOW ^a^^,^^b^	OR 2.62 (0.85 to 8.10)	69 per 1000	94 more per 1000 (10 fewer to 307 more)
Number of patients reaching remission according to DAS28–CRP	2519 (4 RCTs)	⊕⊕◯◯ LOW ^a^^,^^b^	OR 3.05 (1.79 to 5.18)	202 per 1000	234 more per 1000 (110 more to 365 more)
Number of patients reaching 20% improvement according to ACR criteria (ACR20)	3617 (6 RCTs)	⊕⊕⊕◯ MODERATE ^a^^,^^b^	OR 2.33 (1.80 to 3.03)	581 per 1000	183 more per 1000 (133 more to 227 more)
Mortality—not measured	–	–	–	–	–
Number of patients with serious side effects	1643 (3 RCTs)	⊕⊕◯◯ LOW ^b^	OR 1.30 (0.75 to 2.25)	32 per 1000	9 more per 1000 (8 fewer to 37 more)
Change in CRP assessed with: mg/L—not measured	–	–	–	–	–

The risk in the intervention group (and its 95% CI) is based on the assumed risk in the comparison group and the relative effect of the intervention (and its 95% CI). CI: confidence interval; OR: odds ratio; MD: mean difference; VAS: visual analogue scale; DAS28-ESR: disease activity score 28 using erythrocyte sedimentation rate; DAS28-CRP: disease activity score 28 using C-reactive protein; CRP: C-reactive protein; ACR: American College of Rheumatology; MTX: methotrexate. Explanations: ^a^ Significant heterogeneity could be explored. ^b^ Treatment history in moderate-to-severe RA population can be different among the trials. GRADE Working Group grades of evidence. High certainty: we are very confident that the true effect lies close to that of the estimate of the effect; moderate certainty: we are moderately confident in the effect estimate: the true effect is likely to be close to the estimate of the effect, but there is a possibility that it is substantially different; low certainty: our confidence in the effect estimate is limited: the true effect may be substantially different from the estimate of the effect; very low certainty: we have very little confidence in the effect estimate: the true effect is likely to be substantially different from the estimate of effect.

## Data Availability

Data collected and extracted into Excel sheets used in our analyses are available upon request from the corresponding author.

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
