# Peer review of "Janus Kinase Inhibitors Improve Disease Activity and Patient-Reported Outcomes in Rheumatoid Arthritis: A Systematic Review and Meta-Analysis of 24,135 Patients"

_ijms, 2022, doi:10.3390/ijms23031246_

Round 1
Reviewer 1 Report
The study try to compare efficacy of JAK inhibitors with csDMARD from the viewpoint of patient.This is a new viewpoint encouraging for research and important for future.
Author Response
Response to reviewer #1:
Thank you for reviewing our paper, and for the supporting comments.
Reviewer 2 Report
This systematic review is interesting and has some positive sides, but it needs further modifications or organizations before consideration for acceptance.
- Firstly, authors should not claim that it is the "first study," as the truth is 'we do not know the reality.' It must be removed from the abstract, discussion, and elsewhere. They should reword and use moderate statements.
- The abstract must be improved to increase its readability. Authors should concisely include their objective, why, and what they found after meta-analysis.
- The introduction has narrowly focussed on the Janus kinase inhibitors, but authors should describe how these drugs act with detailed information. In addition, it should broadly mention how RA happens and how Janus kinase inhibitors work. Why might these be better or not?
- Table-1 is not organized. Authors must improve and simplify the Table. The comment is moderately applicable to other Tables.
- Figure 1 missing.
- Other figures are not clearly visible and readable.
- Figure 5 is very irrelevant. It must be removed or replaced with a suitable one. It is not a narrative review. Authors should focus only their study on their conclusion or discussion figures. They did not describe the mechanisms of RA as shown in Figure 5 in the body of the manuscript but made an irrelevant Figure.
- The discussion does not flow much. It should explain similar or different outcomes of other meta-analyses of Janus kinase inhibitors or other DMARDs.
- The conclusion should be a bit broad.
- The limitations of this review should be improved.
Author Response
Response to reviewer #2:
Thank you for reviewing our paper and for the helpful suggestions. The manuscript was revised, please find our point-by-point responses to your comments below.
- Firstly, authors should not claim that it is the "first study," as the truth is 'we do not know the reality.' It must be removed from the abstract, discussion, and elsewhere. They should reword and use moderate statements.
Thank you for your suggestion, the intention was only to emphasize this perspective focusing on pain. However, we agree with your comment, therefore the respective parts of the text have been rephrased.
- The abstract must be improved to increase its readability. Authors should concisely include their objective, why, and what they found after meta-analysis.
As you suggested, the abstract has been updated, the restructured and rephrased parts are the following:
„Pain, fatigue, and physical activity are major determinants of life-quality in rheumatoid arthritis (RA). Janus kinase (JAK) inhibitors have emerged as effective medications in RA and have been reported to exert direct analgesic effect in addition to reducing joint inflammation. This analysis aims to give an extensive summary of JAK inhibitors especially focusing on pain and patient reported outcomes (PRO). MEDLINE, CENTRAL, Embase, Scopus and Web of Science databases were searched until the 26th of October 2020., and 50 randomized controlled trials including 24,135 adult patients with active RA met the inclusion criteria. JAK inhibitors yielded significantly better results in all 36 outcomes compared to placebo. JAK monotherapy proved to be more effective than methotrexate in 9 from 11 efficacy outcomes. In comparison to biological Disease-Modifying Antirheumatic Drugs, JAK inhibitors show statistical superiority in 13 of the 19 efficacy outcomes. Analgesic effect determined by visual analogue scale and American College of Rheumatology (ACR) 20/50/70 response rates were significantly greater in the JAK group in all comparisons, and no significant difference regarding safety could be explored This meta-analysis gives a comprehensive overview of JAK inhibitors and provides evidence for their superiority in improving PROs and disease activity indices in RA.”
- The introduction has narrowly focussed on the Janus kinase inhibitors, but authors should describe how these drugs act with detailed information. In addition, it should broadly mention how RA happens and how Janus kinase inhibitors work. Why might these be better or not?
In accordance with your suggestion several parts of the introduction section have been modified and new content has been added as follows:
…„Several genetic, epigenetic and environmental risk factors, such as infections, smoking, diet, and specific microbiome etc. can contribute to the abnormal activation of immune system in RA. Different types of adaptive and innate immune cells, such as B and T cells, neutrophils, mast cells, macrophages and dendritic cells, as well as synovial fibroblasts involved in the pathogenesis produce a wide range of inflammatory cytokines (e.g. tumor necrosis factor-α (TNF-α), interleukin (IL)-1β, IL-6, IL-8, IL-12/IL-23, IL-17, IL-18, IL-32 and interferon-γ (IFN-γ)) that are responsible for further enhancing inflammatory processes. [1]
Janus kinase signaling (JAK1-, JAK2-, JAK3- and Tyrosine Kinase 2/ TYK2) linked to the Signal Transducer and Activator of Transcription (STAT) pathway is directly or indirectly associated with most of these cytokines, therefore plays a key role in the inflammatory mechanisms in RA. JAKs are non-receptor tyrosine kinases activated by the binding of certain ligands to their membrane bound receptor. Once activated, JAKs phosphorylate the associated receptor forming a docking site for the members of the STAT protein family. After docking, STAT proteins also get phosphorylated by JAKs leading to their dissociation and dimerization. The activated STAT dimers then translocate to the nucleus and take part in the regulation of transcription. [6] This process can be selectively blocked by small molecule JAK inhibitors [7]. In addition, they have been described to directly inhibit pain processing [8], since the receptors of many inflammatory cytokines regulating through the JAK-STAT pathway are expressed on neuronal and glial cells [8], but these neuronal mechanisms are currently poorly understood.”
- Table-1 is not organized. Authors must improve and simplify the Table. The comment is moderately applicable to other Tables.
Thank you for your suggestion, according to your proposal Table 1 is reformatted and many parts have been removed to increase readability. However, we think this information has additional value for the readers, therefore the original table has been transmitted to the supplementary file. Minor modifications have also been implemented in Table 2, but since this table follows the original format according to the GRADE approach, we preferred to keep its original structure.
- Figure 1 missing.
We are very sorry for the administrative mistake, it was originally included among the files, however during the submission process something must have gone wrong. It is reuploaded in the manuscript, too.
- Other figures are not clearly visible and readable.
Thank you for the comment, the format of the figures has been modified. The fonts were increased and, in some places, reformatted, as well as background color was added to improve readability and visibility.
- Figure 5 is very irrelevant. It must be removed or replaced with a suitable one. It is not a narrative review. Authors should focus only their study on their conclusion or discussion figures. They did not describe the mechanisms of RA as shown in Figure 5 in the body of the manuscript but made an irrelevant Figure.
Thank you for your reflection. The goal of the figure was also to emphasize that the main focus of this article is how JAK inhibitors are connected to pain and consequently other PROs, However, your opinion provides another aspect, therefore Figure 5 was removed from the main text. Since we think it is important to highlight in the dicussion part, it was inserted as a supplementary file.
- The discussion does not flow much. It should explain similar or different outcomes of other meta-analyses of Janus kinase inhibitors or other DMARDs.
Thank you for this suggestion the discussion has been updated now includes the following:
“Previously no other meta-analysis examined the effect of JAK inhibitors on PROs, however some studies included outcomes regarding life-quality. One analysis included the improvement in HAQ-DI and found JAK inhibitors overall as effective as bDMARDs, except for tofacitinib 10 mg showing significant advantage compared to adalimumab, and baricitinib 2 mg performing slightly worse than bDMARDs [105]. In our analysis the different JAK inhibitors were pooled into one group and significant difference between JAK inhibitors versus bDMARDs were not observed either. SF-36 questionnaire exploring mental health was also examined in some studies. One of them found JAK inhibitors slightly more effective than bDMARDs [106], however another study revealed no significant difference, similarly to our results [107].
Importantly, the proportion of patients reaching remission assessed by DAS28-CRP was significantly higher in the JAK group than in case of bDMARDs, which is consistent with the findings of a previous meta-analysis demonstrating the superiority of upadacitinib compared to adalimumab [105]. However, no statistically significant differences were found in remission rates according to DAS28-ESR, SDAI, and CDAI. This difference might be explained by the observation that DAS28-CRP can underestimate the disease activity [108].
Our analysis also demonstrated the superiority of JAK inhibitors compared to bDMARDs in several other efficacy outcomes e.g. ACR20/50/70, in line with previous meta-analyses also describing the advantages of tofacitinib and baricitinib over adalimumab regarding efficacy outcomes. [109-111] Although this difference was in most outcomes, statistically significant, the clinical relevance of the individual results might seem uncertain. JAK inhibitors have other advantages compared to bDMARDs as well. bDMARDs tend to lose their efficacy with time, which might partially be explained by the formation of anti-drug antibodies [112]. In addition, JAK inhibitors are small molecules appropriate for oral administration, which makes patients’ compliance better. Therefore, our results together with other advantageous features of JAK inhibitors, clearly suggest their benefits over bDMARDs.
MTX is usually the initial treatment option in RA [3], but in many patients the therapy with MTX alone is not sufficient to control the disease and the addition of a b/tsDMARD often needed [113]. However, several studies also examine the difference between JAK inhibitor monotherapy and MTX in MTX-naïve population, suggesting the possibility of an early administered JAK inhibitor monotherapy. Our results confirm and extend the findings of a previous network meta-analysis, demonstrating the significant advantage of different JAK-inhibitors in ACR 50 and 70 response compared to MTX alone [12]. We found the superiority of JAK inhibitor monotherapy compared to MTX in all investigated outcomes, except for the number of patients in remission defined by DAS-28 ESR and SDAI. On the other hand, significantly more patients reached remission thresholds defined by DAS-28 CRP and CDAI as well as achieved ACR20/50/70 response rates in the JAK inhibitor group, clearly supporting the overall efficiency of JAK inhibitor monotherapy compared to MTX, especially in contrast to bDMARDs, which seem more advantageous in combination therapy. [114]”
- The conclusion should be a bit broad.
The conclusion has been extended, as you recommended:
„Pain is one of the most crucial symptoms for RA patients leading to impaired quality of life, motility and working capability. This comprehensive and detailed systematic review and meta-analysis provided evidence for the efficacy and safety of JAK inhibitors with specific emphasis on pain and other PROs. They proved to be highly effective in all investigated outcomes in comparison with placebo, without raising any safety concerns. In contrast to bDMARDs, JAK inhibitors showed significant advantages not just in improving quality of life, but also reducing inflammation in patients with inadequate response to at least one csDMARD or bDMARD. Furthermore, we also proved their significant efficacy in reducing disease activity over MTX monotherapy. Future studies investigating the long-term differences of JAK inhibitors in comparison to other medications, as well as head-to-head trials of the different JAK inhibitors would be important to further evaluate the different aspects of their effectiveness.
- The limitations of this review should be improved.
Thank you for your suggestion, the limitation section has been thoroughly revised as follows:
“Our analysis has some limitations. Firstly, most of the included trials did not exceed six months or switched therapy beforehand, therefore, these results should be interpreted only for short-term, since the long-term effect of JAK inhibitors could not be observed effectively. The completion of the currently ongoing follow-up studies as well as newly conducted analyses would be necessary to draw a clearer conclusion. Secondly, considerable statistical heterogeneity appeared in some of our analyses, which can be explained by evaluating JAK inhibitors together. On the other hand, although the demographic characteristics of the included patients in the RCTs, the study designs or the different types of the permitted concomitant therapy were in general comparable, some smaller differences could be observed, which could also be responsible for the heterogeneity. However, the c2 test is capable of detecting even a small amount of heterogeneity in case of a meta-analysis including a large amount of RCTs just as in our analysis, without necessarily indicating clinically relevant heterogeneity [115]. Besides, secondary analyses also helped to find cause of heterogeneity in some comparisons. Finally, most of the RCTs included patients with moderate-to-severe RA. Although, this is in line with the clinical practice, where JAK inhibitors are administered when remission or low disease activity cannot be reached [3], our findings should be interpreted with caution.”
Round 2
Reviewer 2 Report
Authors made significant improvement.